# Cold Drinking Water Boosts the Cellular and Humoral Immunity in Heat-Exposed Laying Hens

**DOI:** 10.3390/ani13040580

**Published:** 2023-02-07

**Authors:** Hatem M. Eltahan, Chang W. Kang, Vishwajit S. Chowdhury, Hossam M. Eltahan, Mostafa A. Abdel-Maksoud, Ayman Mubarak, Chun Ik Lim

**Affiliations:** 1Animal Production Research Institute, Agriculture Research Center, Agriculture Ministry, Sakha, Kafr El-Sheikh 33717, Egypt; 2Postdoc at the Department of Animal Science, Jeonbuk National University, Jeonju 54896, Republic of Korea; 3College of Veterinary Medicine, Jeonbuk National University, Jeonju 54596, Republic of Korea; 4Division for Experimental Natural Science, Faculty of Arts and Science, Graduate School of Bioresource and Bioenvironmental Science, Kyushu University, Fukuoka 819-0395, Japan; 5Botany and Microbiology Department, College of Science, King Saud University, Riyadh 11451, Saudi Arabia; 6Poultry Research Institute, National Institute of Animal Science, RDA, Pyeongchang 25342, Republic of Korea

**Keywords:** cold drinking water, laying hens, heat stress, immunity, cytokines

## Abstract

**Simple Summary:**

The current study shows that using cold water under high ambient temperature (CW: 15 ± 1 °C; HT: HT: 35 ± 1 °C) in heat-exposed laying hens is capable of maintaining productive efficiency and immune-suppressing under heat stress. The feed intake and egg production were enhanced after using the cold water under heat stress. Moreover, the cold water restored the decline in the level of B-cell, helper T cells, and the ratio of helper/cytotoxic T cells in peripheral blood mononuclear cells, as well as the concentration of IL-2, IFN-γ, and immunoglobulin G in plasma. Therefore, cold water is one of the mechanisms that can be considered under heat stress.

**Abstract:**

This study aimed to investigate the effects of cold drinking water on cellular and humoral immunity in heat-exposed laying hens. One hundred and eight laying hens at 19 weeks old were placed into three treatments with six replicates of six hens in each group as follows: (1) hens were provided with normal drinking water (NW) under the control of thermoneutral temperature (CT: 25 ± 1 °C; CT + NW), (2) hens were provided with NW under high ambient temperature (HT: 35 ± 1 °C; HT + NW) for 8 h/d for a month, and (3) hens were treated under HT with cold drinking water (CW: 15 ± 1 °C; HT + CW) for 8 h/d for a 4-weeks. Then, the feed consumption, egg production, egg weight, feed conversion ratio, and blood immune parameters were investigated. The results showed that cold drinking water (CW) caused a significant (*p* < 0.05) recovery in the reduction of food intake and egg production due to heat stress; however, there was no significant effect (*p* > 0.05) on egg weight and feed conversion ratio. Moreover, CW significantly (*p* < 0.05) restored the immune-suppressing effects of heat stress on the contents of peripheral blood mononuclear cells, including B-cell (BU-Ia), helper T cell (CD4), and the ratio of helper/cytotoxic T cell (CD4/CD8). In addition, CW significantly (*p* < 0.05) recovered the reduction on the level of mRNA expression of interleukin-2 (IL-2) and interferon-gamma (IFN-γ), as well as significantly (*p* < 0.05) restored the reduction of plasma concentration of IL-2, IFN-γ and immunoglobulin G in heat-stressed laying hens. These results prove that CW increased heat dissipation and enhanced feed intake, egg production, and cellular and humoral immunity in heat-exposed laying hens.

## 1. Introduction

High ambient temperature (HT) in the summer season is one of the major environmental challenges facing the poultry industry worldwide. Commercial chickens for intensive meat and egg production are more sensitive to diseases due to a reduction in immune response throughout genetic selection improvement [1,2]. Heat stress harms all age groups of chickens, including young chicks [3], broilers of market age [4], and adult layers [5]. Several studies have demonstrated that heat stress negatively impacts feed intake, body weight, behaviours, egg production, eggshell quality, gut integrity, immunity, and mortality [6,7,8,9]. It has been reported that high temperature triggers the hypothalamus’s appetite centre to send nerve impulses that inhibit feed intake [10]. Moreover, hens decrease their daily feed intake by roughly 1% to 1.5% for every degree over the thermoneutral zone (20 to 25 °C) [11].

Furthermore, numerous studies have indicated that heat stress has immunosuppressive effects in laying hens. The immune system of chickens consists of innate immunity, nonspecific lymphocytes (such as a macrophage, neutrophil, natural killer cell, or dendritic cell), and adaptive immunity, including cellular (T cells) and humoral (B cells) immunity. B and T cells are the white blood cells made in the bone marrow. B cells mature in the bone marrow, while the T cells travel to the thymus and mature there. The humoral B cells can transform into plasmocytes helping signal plasma cells to create three different types of antibodies called immunoglobulin G, M, and A (IgG, M, and A, respectively), which effectively coat or kill particular microorganisms. Furthermore, the cellular (T cells) and humoral (B cells) interact together via chemical messengers known as cytokines, which include interleukin-2 (IL-2) and interferon-gamma (IFN-γ), as well as many other components [12]. In addition, IL-2 has been shown to boost T cell proliferation and IFN-γ to improve adaptive cellular immunity [13].

It has been demonstrated that heat stress reduces innate and adaptive immunity in hens, as measured through phagocytic response and serum antibody titers [14,15,16]. Heat stress, for example, lowered bursal weight and the number of lymphocytes inside the bursal cortex and medulla sections, as well as the number of circulating lymphocytes, and increased the number of heterophils [8]. Moreover, heat stress reduced the relative weights of the spleen and thymus [17,18], declined the liver weights [19,20], decreased the response of systemic humoral immunity [9], and reduced the lymphocytes and IgA-secreting cells in the intestinal tract [6]. Therefore, it is critical to discover beneficial methods and techniques to alleviate heat stress-induced adverse effects on immunological functioning in laying hens.

Water is one of the nutrients with several positive benefits, making it the most crucial and essential nutrient for overall health. Water aids in the regulation of body temperature, feed digestion and absorption, nutrient transfer, and waste disposal from the body [21]. Usually, under average normal ambient temperature, the birds consume water approximately 1.6 to 2.0 times more than feed based on their weight [22]. During heat stress, water consumption quadrupled to increase heat dissipation via convection, conduction, and radiation [23]. According to NRC, water consumption rises roughly 7% for every 1 °C increase in ambient temperature over 21 °C in broilers [24].

Cold drinking water (CW) has been shown to maintain high feed intake and egg production in hens under heat stress (30 °C) [25,26]. On the other hand, CW immersions in the human body three times a week for six weeks increased the total T lymphocytes (CD3), helper T cells (CD4), and B lymphocytes while suppressing cytotoxic T cells (CD8) [27]. Furthermore, Tipton et al. [28] demonstrated the advantages of using cold water for heat dissipation and immunological enhancement in humans. However, it is yet unknown whether CW has any protective effect on laying hen immunity when exposed to heat stress.

In this study, we examined how cold water affects the activity of immune cells, such as cellular immunity (CD3, CD4, CD8, IL-2, and IFN-γ) and humoral immunity (B-cell and IgG), in laying hens.

## 2. Materials and Methods

### 2.1. Animal Housing and Management

The layers were housed in metal wire cages randomly. The dimensions of each cage were 60 × 60 × 40 cm (2 layers/cage). The cages were divided into two-block chambers (4 × 4.2 × 2.6 m) for a control thermoneutral temperature (CT: 25 ± 1 °C) and HT (35 ± 1 °C) for eight h/day (h/d). Each birdcage was equipped with a feeding and drinking station. For the cold drinking water group, water at 15.0 ± 1 °C was passed through the chiller directly to the cup and nipple waterer as described elsewhere [29]. The present study was conducted following the guidelines for animal experiments at the Faculty of Agriculture of Jeonbuk National University and followed the ethical approvement number 2021-0168.

### 2.2. Experimental Design

A total of one hundred and eight 19-week-old Hy-line brown layers weighing 1.4 to 1.8 kg were divided into three treatment groups with six replicates of six hens in each group. The first group was treated with normal drinking water (NW) under CT (25 ± 1 °C; CT + NW). Then, the second group was given NW under HT (35 ± 1 °C) for 8 h/d for a month (HT + NW). Finally, the third group was treated with cold drinking water (15 ± 1 °C, CW) under HT for 8 h/d for a month (HT + CW). For four weeks, the CW treatment was paired with HT treatment for 8 h daily from 11 a.m. to 7 p.m., then kept on the NW until the next day. The hens were placed for adaptation in the experimental chambers two weeks before the experiment started. The hens were exposed to 17 h of light per day, and the humidity was 60% during the investigation, while water and commercial feed, according to [25] (Table 1), were available ad libitum.

### 2.3. Data and Sample Collection

The feed was provided twice daily. The feed intake was measured by collecting and weighing the leftovers in the feeder and calculating the average feed intake for each weekly interval in replications, while daily water intake was determined by measuring water on the morning of the first day and weighing it back the next morning. Moreover, eggs were collected and recorded daily at 4 p.m. Weekly eggs were pooled and analysed related to the number of hens for each cage in each replication. The feed conversion ratio was calculated using weekly feed consumption divided by the weekly egg mass for the four weeks, while egg mass per hen per day was calculated as the average egg production percentage multiplied by the average daily egg weight. Furthermore, blood samples were collected before the experiment and after the 15th and 30th days of the trial to examine the immune cells in peripheral blood mononuclear cells (PBMC).

### 2.4. Isolation of Plasma and PBMC

Fifteen ml of chicken blood was obtained from ten hens of each treatment in a sterile glass tube containing 150 IU of preservative-free heparin (Sigma, St Louis, MO, USA). Plasma and PBMCs consisting of lymphocytes, monocytes, and polymorphs were isolated with the standard Ficoll-Hypaque (Histopaque, Sigma) density gradient centrifugation method to the manufacturer’s instructions. One aliquot of 5 × 10^6^ cells was used immediately for RNA extraction, and the remaining (usually ≥ 10 × 10^6^) cells were used to determine the concentration of cytokines content.

### 2.5. Isolation of PBMC Total RNA and Quantitative Real-Time PCR

Total RNA was extracted from the chicken PBMCs using the modified guanidinium thiocyanate-phenol-chloroform method, according to Gauthier et al. [30]. According to the manufacturer’s instructions, cDNA was synthesized using 1 μg of total RNA and the PrimeScript® RT reagent Kit with gDNA Eraser (Takara, Shiga, Japan). All primers were evaluated using routine PCR and gel electrophoresis before real-time PCR (TaKaRa PCR Thermal Cycler Dices, Takara, Shiga, Japan). The expression of chicken IL-2 and IFN-γ in the PBMC were quantified with real-time PCR following the steps that are written elsewhere in Eltahan et al. [3]. The primer sequences are presented in Table 2. Relative mRNA expressions have been calculated by comparing the thermal cycles needed to generate threshold amounts of product (PCR-ct). PCR-ct was calculated for the chicken IL-2 and IFN-γ, as well as the chicken RNA polymerase-II (RP-II) as a housekeeping gene, since it was confirmed that the RP-II expression level was not altered under the current experimental conditions. The IL-2 and IFN-γ mRNA expression were calculated as 2- ΔΔ PCR-ct, as described elsewhere by Schmittgen and Livak [31].

### 2.6. Statistical Analysis

The study data were subjected to compare the effects of providing cold drinking water and normal drinking water on the productivity of laying hens under average or HT using one-way analysis of variance (ANOVA) software (v 9.4; SAS Institute, 2016, Gary, CA, USA) [32]. We chose *p* < 0.05 as the minimum acceptable significance level except regarding the blood analysis. Results are shown as mean, standard deviation, and coefficient of variance.

## 3. Results

### 3.1. Production Performance Following CW Administration under Heat Stress Exposure in Hens

The egg production and feed intake were significantly (*p* < 0.05) reduced under HT+NW compared with CT+NW in laying hens (Table 2). However, the CW of 15 ± 1 °C significantly (*p* < 0.05) improved the adverse effects of HT on egg production and feed intake. In contrast, water intake increased from HT under the NW and CW compared with CT. In addition, there was no significant difference between the HT or CW on the egg weight and feed conversion ratio (Table 3).

### 3.2. The Contents of PBMCs Following CW Administration under Heat Stress Exposure in Hens

The percentage of B cells significantly (*p* < 0.05) declined after exposure to the HT for 15 and 30 days in laying hens (Table 4). However, we found that CW treatment (HT + CW) significantly eliminated the adverse effect of HT on the number of B cell lymphocytes after 15 and 30 days in heat-stressed laying hens. In addition, the number of CD4 and the ratio between CD4/CD8 have shown the same significant (*p* < 0.05) pattern of the reduction under heat stress treatment (HT + NW). This reduction due to HT exposure was impeded significantly (*p* < 0.05) by the CW treatment (HT + CW) in heat-stressed laying hens after 15 and 30 days (Table 4). In contrast, there were no significant differences in the percentage of CD3 and CD8 in PBMCs in connection to the HT exposure or CW treatments (Table 4).

### 3.3. The mRNA Expression of IL-2 and IFN-γ in PBMCs and Concentrations of Plasma Immune Parameters Following CW Administration under Heat Stress Exposure in Hens

The mRNA expressions of IL-2 and IFN-γ in PBMC significantly (*p* < 0.05) declined after 15 and 30 days of heat exposure in hens under the HT + NW group. However, the CW treatment under HT (HT + CW) significantly (*p* < 0.05) recovered the heat stress-induced decline of the mRNA expressions of IL-2 and IFN-γ in PBMC after 15 and 30 days of heat exposure in hens (Table 5). The levels of IL-2, IFN-γ, and IgG concentrations in the plasma significantly (*p* < 0.05) declined after 30 days of heat exposure (HT+ NW) in hens (Table 5). The analysis has shown that CW treatment (HT + CW) showed a tendency (*p* = 0.09) to restore the adverse high-temperature effects on the concentration of IL-2, IFN-γ, and IgG in plasma after 15 days of administration. It was interesting to note that CW treatment significantly (*p* < 0.05) recovered the adverse effect of HT on the concentration level of plasma IL-2, IFN-γ, and IgG after 30 days of exposure to heat stress in hens (Table 5).

## 4. Discussion

In this study, we demonstrated that the CW could overcome the detrimental effects of summer heat stress on the hen’s productivity and immune system, including CD3, CD4, CD8, IL-2, and IFN-γ in PBMCs in the plasma.

It has been elucidated that heat stress reduced the feed intake as a starting point for the detrimental effects on productivity. It caused a decline in body weight, feed conversion ratio, egg production, egg quality, and immune response in laying hens [33]. These findings are matching with our current study results (Table 3), indicating that exposure to the HT significantly (*p* < 0.05) decreased the feed intake, which might be the result of suppressed metabolic heat production under heat stress. Furthermore, we found a reduction in egg production due to exposure to HT that showed similarity with several studies on laying hens [6,34]. According to Guterrez et al. [25], chilled water (16 ± 0.5 °C) increased feed intake, which reflects increased calcium intake and, consequently, egg production, compared to un-chilled water (23 ± 2.5 °C) under constant temperature at 30 °C in the laying hens. We observed that CW (15 ± 1 °C) significantly (*p* < 0.05) recovered the detrimental effects of heat stress on feed intake and egg production (Table 3). Moreover, Lim et al. [35] reinforced the hypothesis that CW significantly (*p* < 0.05) raised the egg production in Hy-line layer hens under heat stress.

It has been observed that broilers exposed to acute heat stress reduce their feed intake [36] and increase water intake to efficiently control their body temperature with evaporative cooling (panting) [37]. We found that HT increases the water intake and reduces feed intake while using CW slightly declines water intake and consequently increases feed intake. Therefore, cold water under heat stress treatment might reduce the core body temperature through conduction and convection, which could increase feed intake and, consequently, egg production. It will be interesting to investigate the body temperature and the appetite hormone signalling in future research to explain the improvement of egg production after using cold water under heat stress. In contrast, previous research [5,38] indicated a decline in egg weight and feed conversion ratio in heat-stressed laying hens. We found no significant change in egg weight and feed conversion ratio under HT or CW treatment in heat-stressed laying hens. This could be due to our research’s short duration of one month.

Previously, several studies have demonstrated that heat stress has an immunosuppressing impact on laying hens. In agreement with Aengwanich [8], who reported that heat stress decreased bursal weight (the organ for B lymphocyte development, proliferation, and differentiation in chicken), we discovered that heat stress dramatically decreased the percentage of B cells in PBMC after 15 and 30 days of exposure to HT (Table 4). Moreover, Oznurlu et al. [39] have reported that embryonic or post-hatch heat stress leads to bursal follicular atrophy, which could affect B cell production. Another study found that B cells were reduced in both bursal and blood under heat stress [40]. The immunosuppressive status of layers after exposure to heat stress in the current investigation was highlighted by the same pattern of reduction in helper T cells (CD4) and the ratio between helper to cytotoxic T cells (CD4/CD8) in PBMC after 15 and 30 days of exposure to heat stress. Our results have similarities with the previous studies [41,42], which have reported that there had been a reduction in the number of circulating lymphocytes in laying hens under heat stress. Our results showed that CW recovered the harmful effect of HT on declining the percentage of BU-Ia, CD4, and CD4/CD8 in PBMC after 15 and 30 days (Table 4). These health advantages are believed to be a consequence of the physiological and biochemical responses that occur from drinking CW, such as the body’s heat loss through conduction which decreases tissue temperature and, hence, reduces the metabolic rate and oxygen requirement of this cooled tissue [43].

Moreover, we found that gene expression of IL-2 and IFN-γ in PBMC significantly decreased after 15 and 30 days of exposure to HT (Table 5), which provides additional evidence on the immune-suppressing effects of heat stress in laying hens. In addition, our results have shown that CW can restore the reduction of gene expression IL-2 and IFN-γ in PBMC (Table 5) patterns under heat exposure and enhance the laying hen’s immunity. The current results regarding the concentration of cytokines in plasma have strengthened our hypothesis about CW improving the recovery of the detrimental effects of heat stress (Table 5) on the concentration of cytokines IL-2, IFN-γ, and IgG after 30 days but not on 15 days in laying hens’ plasma.

## 5. Conclusions

Our observations suggest that using CW effectively contributed to overcoming the adverse heat stress effects in the decline of the immune cell contents, including cellular immunity CD4, CD8, IL-2, and IFN-γ and humoral immunity of BU-Ia and IgG in PBMC and plasma in heat-stressed laying hens.

## Figures and Tables

**Table 1 animals-13-00580-t001:** The ingredient and nutrient composition of experimental diets.

**Ingredient (%)**	
Corn	55.700
Hard Red Winter Wheat	3.820
Wheat bran	10.220
Soybean meal (48%)	19.370
Monocalcium phosphate (Ca 18%, *p* 21%)	0.830
Limestone (Ca 38.5%)	9.460
Iodized salt	0.300
DL-methionine (99%)	0.100
Vitamin premix ^1^	0.100
Mineral premix ^2^	0.100
Total	100
**Calculated nutrient composition**	
Metabolizable energy (kcal/kg)	2750
Crude protein (%)	16.00
Calcium (%)	4.00
Total phosphorus (%)	0.69
Available phosphorus (%)	0.40
Lysine (%)	0.80
Methionine (%)	0.36
Cysteine (%)	0.29
Arginine (%)	0.99

^1^ Vitamin supplement provided per kilogram of diet: vitamin A, 10,000 IU; vitamin D3, 2500 IU; vitamin E, 20 IU; vitamin B1 1.5 mg; vitamin B2 5.0 mg; vitamin B6, 0.15 mg; vitamin B12 15.0 mg; choline, 300 mg; pantothenate, 12 mg; nicotinic acid, 50 mg; biotin, 0.15 mg; folic acid, 1.5 mg. ^2^ Mineral supplemented provided per kilogram of diet: Fe, 60 mg, Cu,10 mg; Zn 80 mg; Mn, 110 mg; Iodine, 0.48 mg; Se, 0.40 mg.

**Table 2 animals-13-00580-t002:** Primers used for real-time PCR.

Gene	Accession No.	Sequences 5′-3′ (Forward/Reverse)	Annealing Temperature (°C)	Product Size (bp)
INF-γ	NM_205149.2	5′-TGTAGCTGACGGTGGACCT-3′/ 5′-ATGTGTTTGATGTGCGGCTT-3′	60	147
IL-2	NM_204153.2	5′-ACTCTGCAGTGTTACCTGGG-3′/ 5′-CCGGTGTGATTTAGACCCGT-3′	60	140
RP-II	NM_001006448.2	5′-CGACGGTTTGATTGCACCTG-3′/ 5′-CAATGCCAGTCTCGCTAGTTC-3′	64	161

Primers were designed with Primer-Blast (http://www.ncbi.nlm.nih.gov/tools/primer-blast) accessed on 23 November 2021 for interleukins-2 (IL-2), while accessed on 28 October 2021 for interferon-gamma (INF-γ) and accessed on 23 September 2021 for RNA polymerase II (RP-II). According to Eltahan et al. [3], RP-II has been selected as an internal control gene constantly expressed during heat stress.

**Table 3 animals-13-00580-t003:** The effect of cold drinking water on the performance of laying hens after heat stress exposure.

Treatments	Egg Production (%)	Egg Weight (g)	Daily Feed Intake (g)	Daily Water Intake (mL)	Feed Conversion (%)
CT + NW	91.17 ± 1.04 ^a^	62.75 ± 0.39	108.93 ± 0.62 ^a^	217 ± 24.0 ^b^	1.90 ± 0.02
HT + NW	85.07 ± 1.50 ^b^	62.21 ± 0.61	102.95 ± 1.03 ^b^	355 ± 73.8 ^a^	1.95 ± 0.04
HT + CW	89.11 ± 0.66 ^a^	61.84 ± 0.71	107.81 ± 0.39 ^a^	304 ± 63.8 ^a^	1.96 ± 0.02
*p*-value	0.04	0.583	0.045	0.047	0.876

Values are means ± SEM in the number of 24 laying chickens used in each treatment. Different superscripts in the same column indicate significant differences at *p* < 0.05 between the treatments. CT + NW, control thermoneutral temperature (25 ± 1 °C) with regular tap water treatment (25 ± 1 °C); HT + NW, high temperature (35 ± 1 °C) with regular tap water treatment (25 ± 1 °C); HT + CW, high temperature (35 ± 1 °C) with cold water treatment (15 ± 1 °C).

**Table 4 animals-13-00580-t004:** Effects of cold drinking water on the contents of PBMC lymphocyte subsets in laying hens under heat stress.

Period and Treatments	BU-Ia (%)	CD3 (%)	CD4 (%)	CD8 (%)	CD4/CD8
**0 day**					
CT + NW	2.76 ± 0.43	11.61 ± 2.23	48.80 ± 7.64	23.08 ± 3.43	2.12 ± 0.36
HT + NW	2.80 ± 0.35	13.60 ± 2.28	48.65 ± 6.57	23.60 ± 4.97	2.17 ± 0.28
HT + CW	2.67 ± 0.45	14.95 ± 2.49	47.35 ± 9.12	23.30 ± 5.53	2.07 ± 0.23
*p*-value	0.583	0.656	0.523	0.324	0.243
**15 days**					
CT + NW	3.32 ± 0.46 ^a^	13.38 ± 1.39	63.52 ± 7.87 ^a^	21.55 ± 4.66	3.03 ± 0.35 ^a^
HT + NW	2.56 ± 0.41 ^b^	15.73 ± 3.94	52.96 ± 5.25 ^b^	21.21 ± 5.59	2.51 ± 0.25 ^b^
HT + CW	4.23 ± 0.52 ^a^	15.67 ± 2.40	62.00 ± 7.77 ^a^	21.77 ± 4.00	2.95 ± 0.29 ^a^
*p*-value	0.038	0.234	0.037	0.891	0.048
**30 days**					
CT + NW	3.98 ± 0.48 ^a^	15.66 ± 4.88	63.21 ± 7.58 ^a^	16.76 ± 5.03	3.95 ± 0.32 ^a^
HT + NW	2.04 ± 0.34 ^b^	13.68 ± 2.83	47.27 ± 7.29 ^b^	16.12 ± 3.54	2.93 ± 0.21 ^b^
HT + CW	4.31 ± 0.52 ^a^	14.03 ± 2.54	63.80 ± 6.78 ^a^	16.76 ± 3.34	4.05 ± 0.34 ^a^
*p*-value	0.026	0.675	0.027	0.853	0.046

Values are means ± SEM in the number of 10 laying chickens used in each group. Different superscripts in the same column indicate significant differences at *p* < 0.05 between the treatments in the same period. CT + NW, thermoneutral control temperature (25 ± 1 °C) with regular tap water treatment (25 ± 1 °C); HT + NW, high temperature (35 ± 1 °C) with standard tap water treatment (25 ± 1 °C); HT + CW, high temperature (35 ± 1 °C) with cold water treatment (15 ± 1 °C); PBMC, peripheral blood mononuclear cells; BU-Ia, B-cells; CD3, number of lymphocytes in PBMC; CD4, helper T cells; CD8, cytotoxic T cells; PBMC, peripheral blood mononuclear cells.

**Table 5 animals-13-00580-t005:** Effects of cold drinking water on the mRNA expression of cytokines in PBMC and immune parameter concentration in plasma of laying hens after heat stress exposure.

Period and Treatments	The Expression of Cytokines mRNA in PBMC		The Concentration of Cytokines and IgG in Plasma		
	IL-2	IFN-γ	IL-2 (ng/mL)	IFN-γ (ng/mL)	IgG (mg/mL)
**0 day**					
CT + NW	1.01 ± 0.23	0.99 ± 0.31	2.45 ± 0.62	1.15 ± 0.31	3.84 ± 0.56
HT + NW	1.08 ± 0.18	0.98 ± 0.22	2.54 ± 0.59	1.01 ± 0.22	3.72 ± 0.63
HT + CW	1.14 ± 0.32	1.12 ± 0.36	2.45 ± 0.69	1.12 ± 0.27	3.79 ± 0.73
*p*-value	0.983	0.354	0.845	0.987	0.998
**15 days**					
CT + NW	4.46 ± 0.82 ^a^	7.31 ± 1.45 ^a^	2.84 ± 0.55	1.31 ± 0.49	4.06 ± 0.77
HT + NW	2.84 ± 0.55 ^b^	3.59 ± 0.99 ^b^	1.95 ± 0.48	0.83 ± 0.29	2.76 ± 0.69
HT + CW	4.43 ± 0.71 ^a^	6.33 ± 1.32 ^a^	2.36 ± 0.54	1.05 ± 0.46	3.69 ± 0.92
*p*-value	0.038	0.024	0.278	0.144	0.068
**30 days**					
CT + NW	5.44 ± 0.97 ^a^	5.19 ± 1.02 ^a^	2.64 ± 0.67 ^a^	1.79 ± 0.43 ^a^	4.18 ± 0.79 ^a^
HT + NW	1.34 ± 0.63 ^b^	1.44 ± 0.77 ^b^	0.78 ± 0.63 ^b^	0.37 ± 0.19 ^b^	1.71 ± 0.61 ^b^
HT + CW	4.12 ± 0.86 ^a^	5.03 ± 1.22 ^a^	2.31 ± 0.76 ^a^	1.87 ± 0.40 ^a^	3.52 ± 0.73 ^a^
*p*-value	0.024	0.029	0.047	0.049	0.038

Values are means ± SEM in the number of 10 laying chickens used in each group. Different superscripts in the same column indicate significant differences at *p* < 0.05 between the treatments in the same period. CT + NW, thermoneutral control temperature (25 ± 1 °C) with regular tap water treatment (25 ± 1 °C). HT + NW, high temperature (35 ± 1 °C) with standard tap water treatment (25 ± 1 °C). HT + CW, high temperature (35 ± 1 °C) with cold water treatment (15 ± 1 °C); IL-2, interleukins-2; IFN-γ, interferon-gamma; IgG, immunoglobulin G; PBMC, peripheral blood mononuclear cells.

## Data Availability

On fair request, the corresponding author will provide data that support the study’s conclusions.

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
