# Peer review of "Cold Drinking Water Boosts the Cellular and Humoral Immunity in Heat-Exposed Laying Hens"

_animals, 2023, doi:10.3390/ani13040580_

Round 1
Reviewer 1 Report
Dear Authors
Regarding the manuscript title:
Cold drinking water boosts the cellular and humoral immunity 2 in heat-exposed laying hens 3.
The scientific background of the topic was well mentioned in the introduction part. The experiment design quite good, as well as the replicates and methods used, were quite good. The results obtained were presented in tables well discussed with other author’s results. However, there are some observation in the present paper should be corrected and add to improve the quality of the paper.
1- Are you measured water consumption, add the results if you measured it.
2- You started the experiment at 123 day old then kept 14 day for adaptation then started the experiment and the experiment period 30 day, totally 197 day? It is true? If it is true how can egg production obtained 91%?
3- Are you prepared the diets according Hy-line brown layers catalog? It is the production phase 1 the cp must be 18%?
4- Wheat HRW in table. 1 , add full name.
5- How can you calculate FCR in table 1 add method in Materials and methods?
6- Discussion; Add explanation why egg production was increased by CW?
Author Response
Response to Reviewer 1 Comments
We are grateful to the reviewer for the critical comments and useful suggestions that have helped us to improve our paper. As indicated in the following responses, we have considered all comments and suggestions in this revised paper version. Please see a point-by-point response:
Point 1: Are you measured water consumption, add the results if you measured it.
Response 1: Thank you for the thoughtful comment.
Please see lines (255-262); we have written that It has been observed that broilers exposed to acute heat stress reduce their feed intake [36] and increase water intake to efficiently control their body temperature by evaporative cooling (panting) [37]. We found that HT increases the water intake and reduces feed intake, while using CW slightly declines water intake and consequently increases feed intake. Therefore, cold water under heat stress treatment might reduce the core body temperature by conduction and convection, which could increase feed intake and, consequently, egg production. It will be interesting to investigate the body temperature and the appetite hormone signalling in future research to explain the improvement of egg production after using cold water under heat stress. ‘’Thank you for understanding our explanation’’.
Point 2: You started the experiment at 123 day old then kept 14 day for adaptation then started the experiment and the experiment period 30 day, totally 197 day? It is true? If it is true how can egg production obtained 91%?
Response 2: We appreciate your comment on improving our scientific paper. Sorry to confuse it, but we changed it by weeks based on the other respectful reviewer to start at 19 weeks, keep for two weeks for adaptation, and then start the experiment at four weeks. The egg production (%) is the product's mean during the experiment starting from 22-25 weeks. Thank you for understanding our explanation.
Point 3: Are you prepared the diets according Hy-line brown layers catalog? It is the production phase 1 the cp must be 18%?
Response 3: Thank you for your comment on improving our scientific paper. According to Hy-line brown layers catalogue the crud protein ratio should be 17%. However, according to the previous reference, we used the commercial layer diet of 16% CP in the experiment (Guterrez et al., 2009). https://doi.org/10.5713/ajas.2009.80549. Thank you for understanding our explanation.
Point 4: Wheat HRW in table. 1, add full name.
Response 4: Thank you for your comment. We have added the full name in the table
Point 5: How can you calculate FCR in table 1 and add method in Materials and methods?
Response 5: Thank you for your comment. We have added the method accordingly. Please see lines (150-152). We have written that ‘’the feed conversion ratio was calculated using weekly feed consumption divided by the weekly egg mass for the four weeks. While egg mass per hen per day was calculated as the average egg production percentage, multiplied by the average daily egg weight.’’
Point 6: Discussion; Add explanation why egg production was increased by CW?
Response 6: we are glad you commented on improving our research paper. We have written that It has been observed that broilers exposed to acute heat stress reduce their feed intake [36] and increase water intake to efficiently control their body temperature by evaporative cooling (panting) [37]. We found that HT increases the water intake and reduces feed intake, while CW slightly declines water intake and consequently increases feed intake. Therefore, cold water under heat stress treatment might reduce the core body temperature by conduction and convection, which could increase feed intake and, consequently, egg production. It will be interesting to investigate the body temperature and the appetite hormone signalling in future research to explain the improvement of egg production after using cold water under heat stress.

Reviewer 2 Report
The study is interesting, nevertheless a laying hens farm not well equipped for heat stress is not sustainable for an economic point of view.
However, I appreciate the aim of investigating drinking water temperature effect on the different paraemters. I would like to ask why daily water intake was not investigated and another big failure in this study is given by the total absence of investigation about diet and water interaction.
The diet is reported but to me it is necessary put attention on the interaction bewteeen diet factors and water intake. I kindly recommend to review the literature on this aspect and diet factors effect on daily water intake.
Author Response
We are grateful to the reviewer for the critical comments and useful suggestions that have helped us to improve our paper. As indicated in the following responses, we have considered all comments and suggestions in this revised paper version. Please see a point-by-point response:
Point 1: The study is interesting, nevertheless a laying hens farm not well equipped for heat stress is not sustainable for an economic point of view.
Response 1: Thank you for your respectful comment. The current study represents a small scale of how to cope with heat stress during summer in some countries suffering from global warming. We designed it in two blocks chambers for thermoneutral ambient temperature (CT: 25 ± 1°C) and the other one exposed for high ambient temperature HT (35 ± 1°C) for eight h/day.
We believe that a small laying hen farm can provide cold water for the chickens under heat stress. Thank you for understanding our explanation.
Point 2: However, I appreciate the aim of investigating drinking water temperature effect on the different paraemters. I would like to ask why daily water intake was not investigated and another big failure in this study is given by the total absence of investigation about diet and water interaction.
Response 2: Thank you for your thoughtful comment. We have added the water intake results *table 3). As well, please see lines (255-262). We found that HT increases the water intake and reduces feed intake, while using CW slightly declines water intake and consequently increases feed intake. Therefore, cold water under heat stress treatment might reduce the core body temperature by conduction and convection, which could increase feed intake and, consequently, egg production. It will be interesting to investigate the body temperature and the appetite hormone signalling in future research to explain the improvement of egg production after using cold water under heat stress. ‘’Thank you for understanding our explanation’’.
Point 3: The diet is reported but to me it is necessary put attention on the interaction bewteeen diet factors and water intake. I kindly recommend to review the literature on this aspect and diet factors effect on daily water intake.
Response 3: Thank you for your comment on improving our scientific paper. Based on your respectful comments to review the literature on this aspect:
Please see line (77); usually, under average normal ambient temperature, the birds consume water approximately 1.6 to 2.0 times more than feed based on their weight [22]. During heat stress, water consumption quadrupled to increase heat dissipation via convection, conduction, and radiation [23]. As well, please see line (255); we have written that it has been observed that broilers exposed to acute heat stress reduce their feed intake [36] and increase water intake to efficiently control their body temperature by evaporative cooling (panting) [37]. Finally, we demonstrated in line 260 that cold water under heat stress treatment might reduce the core body temperature by conduction and convection, which could increase feed intake and, consequently, egg production. It will be interesting to investigate the body temperature and the appetite hormone signalling in future research to explain the improvement of egg production after using cold water under heat stress. ‘’Thank you for understanding our explanation’’.

Reviewer 3 Report
Interesting read! Attached my further comments.

Author Response
We are grateful to the reviewer for the critical comments and useful suggestions that have helped us to improve our paper. As indicated in the following responses, we have considered all comments and suggestions in this revised paper version. Please see a point-by-point response:
Point 1:
Abstract - Line 17 I recommend using weeks instead of days for laying hens
Response 1: Thank you for your comment on improving our scientific paper. According to your respectful suggestion, we have modified the paper.
Point 2:
Line 43 Reference 6.7 – should be 6,7?
Response 2: Thank you for your suggestion to improve our scientific paper. According to your respectful comment, we have modified it.
Point 3:
-Line 96 I recommend using weeks instead of days,
- Line 101-102 not sure what “no treatment on a remaining day” refers to. Perhaps you could use
other words.
Response 3: We are glad you commented on improving our scientific paper. According to your respectful comments, we have modified the age by the weeks.
Please see lines 99-101; we have changed to be, For four weeks, the CW treatment was paired with HT treatment for 8 hours daily from 11 am to 7 pm, then kept on the NW till the next day. Thank you for understanding our explanation.
Point 4:
What was your experimental design? Did you block by chambers?
Response 4: We appreciate your comments to enhance our scientific paper level. In line 85, we have written that ‘’ The cages were divided into two blocks chambers (4 × 4.2 × 2.6 m) for a control thermoneutral temperature (CT: 25 ± 1°C) and HT (35 ± 1°C) for eight h/day. As well as we have added in line 187 the experimental design ‘’ on-way analysis’’ Thank you for understanding our explanation.
Point 5:
Results
- Line 178 Very interesting results! I recommend writing in passive voice. Instead of saying “...NW
significantly reduced egg production...”, perhaps try “Egg production was reduced in the NW
treatment...”. This is because we may not truly know whether NW + HT reduced egg production
unless data was continuous (i.e first exposing birds to NW + CT) than discrete; there could be
other causes for reduced egg production. Same comment for rest of results section where
applicable.
Response 5: Thank you for your suggestion to enhance our scientific paper. According to your respectful comment, we have modified it. Please find it in lines 192-227.
Point 6:
Discussion
- Line 225 “overcoming” should be “overcome”
Response 6: Thank you for your comment about raising our scientific paper. According to your respectful comment, we have modified it. Please find it in line 238.
Point 7:
Formatting should be consistent with rest of paper (i.e. single spacing, etc.)
Response 7: Thank you for your suggestion to improve our scientific paper. According to your respectful comment, we have modified it. Thank you for your effort.
